# Alternatives for Enhancing Mechanical Properties of Recycled Rubber Seismic Isolators

**DOI:** 10.3390/polym16162258

**Published:** 2024-08-09

**Authors:** Faider S. Rivas-Ordonez, Alex O. Meza-Munoz, Ingrid E. Madera-Sierra, Manuel A. Rojas-Manzano, Edwin D. Patino, Manuel I. Salmerón-Becerra, Shirley J. Dyke

**Affiliations:** 1Departamento de Ingeniería Civil e Industrial, Pontificia Universidad Javeriana Cali, Calle 18 #118-200, Santiago de Cali 760026, Colombia; alexmeza@javerianacali.edu.co (A.O.M.-M.); alejandro.rojas@javerianacali.edu.co (M.A.R.-M.); 2Facultad de Artes Integradas, Escuela de Arquitectura, Departamento de Tecnología de la Construcción, Universidad del Valle, Calle 13, #100-00, Santiago de Cali 760042, Colombia; ingrid.madera@correounivalle.edu.co; 3Lyles School of Civil Engineering, Purdue University, 585 Purdue Mall, West Lafayette, IN 47907-2088, USA; patino7@purdue.edu (E.D.P.); salmeron@purdue.edu (M.I.S.-B.); sdyke@purdue.edu (S.J.D.); 4School of Mechanical Engineering, Purdue University, 585 Purdue Mall, West Lafayette, IN 47907-2088, USA

**Keywords:** recycled rubber, base isolation, fiber-reinforced elastomeric isolator, isolator reinforcement, layer adhesion

## Abstract

Base isolators, traditionally made from natural rubber reinforced with steel sheets (SERIs), mitigate energy during seismic events, but their use in developing countries has been limited due to high cost and weight. To make them more accessible, lighter, cost-effective reinforcement fibers have been utilized. Additionally, the increasing use of natural rubber has caused waste storage and disposal issues, contributing to environmental pollution and disease spread. Exploring recycled rubber matrices as alternatives, this study improves seismic isolators’ mechanical properties through modified reinforcements and layer adhesion. Eight reinforcement materials and eight adhesives, which may be activated with or without heat application, are systematically evaluated. Employing the chosen reinforcements and adhesives, prototypes are tested mechanically to examine their vertical and horizontal performance through cyclic compression and cyclic shear testing. Two innovative devices using recycled rubber matrices were developed, one using a layering technique and another through a monolithic approach shaped with heat and pressure. Both integrate a fiberglass mesh reinforced with epoxy resin; one employs a heat-activated hybrid adhesive, while the other uses a cold bonding adhesive. These prototypes exhibit potential in advancing seismic isolation technology for low-rise buildings in developing countries, highlighting the viability of recycled materials in critical structural applications.

## 1. Introduction

Seismic isolation is a design strategy employed to minimize the damage resulting from seismic action. It works by controlling the energy dissipation produced during earthquakes through specialized seismic isolation devices [1]. Isolation devices often consist of layers of natural rubber combined with rigid steel plate reinforcements called steel elastomeric rubber isolators (SERIs) or flexible reinforcements like fibers (FREIs). The inherent mechanical properties of the rubber layers enable horizontal movement. Simultaneously, the reinforcement provides stability against vertical loads, preventing lateral bulging of the elastomer when subjected to these stresses [2].

Despite their efficacy, seismic isolation devices are not extensively incorporated in contemporary construction. Their application is confined mainly to critical infrastructure such as hospitals and bridges, or establishments with valuable or mission-critical content, such as data centers, communication hubs, high-tech manufacturing facilities, and museums. Many developing nations, where construction practices frequently depend on empirical and traditional knowledge, experience considerable structural damage in seismic events [3,4]. To underscore the urgency of this issue, between 2011 and 2021 in the world, there have been 17,653 earthquakes with magnitudes of 5.0 to 5.9; 1424 with magnitudes of 6.0 to 6.9; 149 with magnitudes of 7.0 to 7.9; and 13 surpassing a magnitude of 8. These seismic events have resulted in over 40,000 casualties worldwide in the past decade alone [5].

The adoption of seismic isolators in developing countries has presented considerable challenges, rooted primarily in economic and technical challenges [6]. Prominent barriers associated with seismic isolators include their substantial cost and weight. The weight of a single isolator can reach or even surpass a ton, presenting significant logistical and engineering obstacles [7]. This excessive weight is primarily attributed to the use of steel plates in the reinforcement process of seismic isolators (SREIs), which implies a prolonged and costly manufacturing process [8].

Consequently, implementing flexible reinforcements in isolators has introduced significant cost and weight reduction improvements, increasing accessibility and ease in the structural construction process [9]. Within this category, fibers emerge as notable elements, each exhibiting specific characteristics that make them suitable for study and application in structural engineering. Common examples of frequently used fibers include glass, carbon, and nylon, chosen due to their mechanical properties, high modulus of elasticity, specific construction applications, and acquisition cost [10]. Specifically, devices with this type of reinforcement are characterized by their increased flexibility and display vertical stiffness like SREIs [11].

FREIs integrate fiber and rubber in their fabrication. A robust adhesion between layers is imperative to optimize the composite material’s performance. The interface bond significantly contributes to the rubber’s fatigue resistance, particularly under shear movements. This interdependence highlights the importance of interfacial bonding in maintaining the integrity and longevity of the composite under dynamic stress conditions. As Zhong et al. [12] indicated, this interaction not only reinforces the material’s cohesive structure but also markedly increases its durability against abrasion and cyclic loads. Historically, an adhesive called Chemlok was the most common solution for this critical bonding in SERIs and FREIs [13,14]. However, its application demands a thermal curing process, which inadvertently extends the processing duration and, as a result, diminishes its practical activity, a challenge highlighted by Hedayati et al. [15].

Given the previously outlined context, researchers have formulated flexible reinforcement and adhesive alternatives to improve the mechanical behavior of these devices. Moon et al. [7] conducted research using steel reinforcements and various types of fibers—namely polyester, nylon, glass, and carbon—embedded within a natural rubber matrix treated with isocyanate and resorcinol-formaldehyde latex (RFL) adhesive. Their results demonstrated that nylon fibers exceeded the vertical stiffness of polyester and glass by approximately 40,000 kg/mm, indicating superior vertical performance suitable for low-rise structures. Notably, carbon fibers demonstrated superior vertical stiffness and damping properties, significantly surpassing steel reinforcement by a factor of three. This work led to the conclusion that carbon fibers achieved markedly better performance than SERIs, with damping capabilities enhanced up to twenty-five times. Similarly, Russo et al. [10] assessed prototypes by altering the configuration of carbon fiber to bidirectional (bd) and quadrilateral (qd) orientations without utilizing any adhesive at the interface with natural rubber. Their research clarified that, relative to bd alignments, qd configurations exhibited a substantial increase in both vertical stiffness and damping, by 30% and 8.35% respectively. This investigation highlighted the quadrilateral configuration’s superior performance in both vertical and horizontal planes within seismic isolators, with a greater energy dissipation capacity than steel reinforcement.

Tan et al. [16] introduced engineering plastic sheets as reinforcements. Composed of an unsaturated polyester resin and fiberglass cloth, these sheets resulted in a vertical stiffness of 320.28 kN/mm and 23.06% damping at a 100% deformation under a vertical compressive stress of 15 MPa. This performance not only proves their efficacy in applications for low-rise buildings but also positions them as superior in damping compared to high-damping isolators, with an 8% increase in damping. Additionally, these engineering plastic sheets present a cost-effective and lighter-weight alternative.

As previously emphasized, carbon fiber has yielded commendable results when used as reinforcement in seismic isolators. However, in developing countries like Colombia, its acquisition can be cost-prohibitive [17]. Addressing this, Madera et al. [18] and Losanno et al. [17] investigated alternatives, integrating steel, carbon, nylon, and polyester fibers with natural rubber using polyurethane adhesive. This aimed to find more affordable solutions than carbon and steel. Their findings showed carbon fiber reduced horizontal stiffness by 24%, improving damping by 6.8% over steel but decreasing vertical stiffness by 33%. Nylon fibers decreased horizontal stiffness by 39% and slightly increased damping by 2.8%, yet vertical stiffness dropped by 57%, making it inadequate for isolators. Notably, Losanno et al. [17] discovered that polyester fiber, more flexible than steel, performed well under horizontal stress with up to 100% deformation and achieved over 20% damping, reducing horizontal stiffness. This behavior makes it more suitable for low-rise buildings at a lower cost, which is beneficial for developing countries.

Moreover, Ortega et al. [19] conducted a mechanical examination of seismic isolator prototypes made from recycled rubber, as part of research on seismic isolators (FREIs) with a matrix of the same material [20], responding to alarming statistics from the International Union for Conservation of Nature that 28% of oceanic plastic waste originates from discarded tires. In Colombia, nearly 950,000 tires are dumped annually in landfills, posing environmental and health threats from hazardous emissions during incineration. Ortega et al. [19] reinforced devices using a bidirectional polyester fiber, developed by Lossano et al [17] and Madera et al. [18], also bonded with polyurethane adhesive [20]. The results indicated a decrease in vertical stiffness of 41% compared to the natural rubber prototypes developed by Losanno et al. [21]. Furthermore, vertical stiffness assessments from cyclic testing showed that due to particle rearrangements within the recycled rubber under compressive forces, the experimental measurements (11.8 kN/mm) were 46% lower than theoretical values (20 kN/mm). In addition, shear tests demonstrated peak displacements of 16 mm, leading to an increase in horizontal stiffness of the prototypes developed by Losanno et al. [17] from 70 kN/m to 600 kN/m, a remarkable increase of 757%. Notably, these tests did not permit the occurrence of roll over, a phenomenon previously documented by the same researchers. This characteristic is a desirable feature in disconnected isolators.

This research introduces improvements to the mechanical properties of seismic isolators by refining reinforcement and adhesive techniques in recycled rubber materials. Utilizing a series of mechanical evaluations, including compression tests, shear adherence tests, and cyclic shear tests on prototypes, the study aims to expand the understanding of base isolator performance with a recycled rubber matrix to enhance vertical and horizontal stiffness and damping of the prototypes developed in other research. This initiative seeks to promote the use of readily available and manageable materials that fulfill essential mechanical criteria, thereby advancing the field of seismic isolation technology.

## 2. Mechanical Evaluation of the Materials of the Seismic Isolator

### 2.1. Theoretical Formulation

In the realm of seismic isolation devices, the assessment of mechanical properties such as vertical and horizontal stiffness becomes indispensable when forces act in dual directions. This evaluation encompasses an analysis of the shear modulus and damping capacities, which are key in confirming the effectiveness of the devices developed in this research under seismic loads. To estimate the properties of the developed prototypes, this study aligns with the methodologies outlined by Kelly and Konstantinidis [22] and extends the investigation initiated by Losanno et al. [21]. A calculation method for determining vertical stiffness is proposed, considering the properties of flexible reinforcement, as demonstrated in Equation (1):(1)$$K_{v}=\frac{E_{f}^{c}A}{Hr}$$
where $E_{f}^{c}$ is the matrix-reinforcement module, A is the isolator cross-section, and Hr is the total elastomer thickness. $E_{f}^{c}$ depends on parameters such as α and β. Here, α is a function of the modulus of elasticity ($E_{f}$), thickness ($t_{r}$), Poisson’s ratio of the reinforcement (v), the thickness of the rubber layer ($t_{r}$), which is calculated from the design of the seismic isolator with the recommendations given by Losano et al. [21], and the device’s radius (R). Meanwhile, β depends on the shear modulus (G), the device’s radius (R), the compressibility factor (K), and the thickness of the rubber sheet ($t_{r}$), as shown in Equations (2) and (3).
(2)$$\alpha^{2}=\frac{12\left(1-v^{2}\right)G_{20}R^{2}}{\left(E_{f}t_{f}t_{r}\right)}$$
(3)$$\beta^{2}=\frac{12G_{20}R^{2}}{Kt_{r}}$$
(4)$$G=\frac{\tau_{max}-\tau_{min}}{\gamma_{smax}-\gamma_{smin}}$$

The shear modulus (*G*) is calculated using Equation (4). Where *γ*_*s**m**a**x*_ and *γ*_*s**m**i**n*_ are the maximum and minimum shear deformations of the hysteresis cycles, respectively; *τ*_*m**a**x*_ and *τ*_*m**i**n*_ are the maximum and minimum shear stresses (*τ* = *F*/*A*). *F* represents the measured force, and *A* is the specimen’s shear area [17].

Furthermore, the damping ratio provides the energy dissipation capacity ($\beta_{d}$) [6]. This depends on the energy dissipated in each cycle $W_{d}$, the effective horizontal stiffness *K_eff_*, and the average of the maximum positive and negative displacements, as presented in Equation (5) [23]:(5)$$\beta_{d}=\frac{W_{d}}{2\pi K_{eff}{{∆}^{2}}_{max}}$$
(6)$$K_{eff}=\frac{F_{max}-F_{min}}{u_{max}-u_{min}}$$
where $K_{eff}$ relies on $F_{max}$ and $F_{min}$, which are the forces at maximum and minimum deformation, and *u_max_* and *u_mín_* represent the maximum and minimum deformations, respectively, as demonstrated in Equation (6). Regarding the horizontal response of Unbonded Seismic Isolators (FREIs), which exhibits nonlinear behavior influenced by vertical load and horizontal displacement, Losanno et al. [21] propose a methodology for horizontal stiffness determination, as depicted in Equation (7).
(7)$$K_{H}=\frac{G_{eff}A_{eff}}{t_{r}}$$
(8)$$\left\{\begin{array}{c} A_{eff}=A,\;for\;d\;\leq \;d_{0}\;\\ A_{eff}=A-A_{d},\;for\;d>d_{0} \end{array}\right.$$

Step 1: The effective shear modulus ($G_{eff}$) is derived from the rubber characterization curve for expected deformation levels. Step 2: The compressed isolator height (h), defined as $h=H-u_{th}$, is analytically estimated. An expression is derived considering theoretical vertical displacement under design load ($u_{th}$) as the sum of the uncompressed rubber displacement ($u_{r}$) and confined rubber displacement ($u_{rf}$), i.e., $u_{th}\;=\;u_{r}+u_{rf}.$ The first part, $u_{r}$, is calculated as the product of the initial height H and unitary deformation ε at the stiffness change limit ($u_{r}=H\varepsilon$). This deformation *ε* is obtained from pure rubber compression test results. The second part, $u_{rf},$ is computed using $u_{rf}=∆P/K_{Vd}$, where (∆P) is the maximum applied load (*P*) minus the load at the stiffness change limit ($P_{sc}$); the theoretical vertical dynamic stiffness $K_{Vd}$ is obtained from Equation (1). Step 3: The effective area is calculated based on the deformation level. For this purpose, the Russo et al. [10] method was modified to apply to a circular isolator. Consequently, the isolator portion in contact with supports and subjected to pure shear ($A_{eff}$) equals the total area (A) minus the detached semicircle area ($A_{d}$) [21]. The separate area is calculated as $A_{d}=R^{2}/2(\theta -\sin\theta )$ with $\theta =2\;\sin^{-1}\left(c/2R\right)$ and the length $c=\sqrt{(R-s/2)8s}$. The separation point ($d_{0}$) and the disconnected portion length (s) are calculated as $d_{0}=\sqrt{H^{2}-h^{2}}$ and $s=d-d_{0}$, respectively. Based on $d_{0}$ and the displacement level (d), $A_{eff}$ is calculated according to Equation (8). Step 4: Finally, $K_{H}$ is calculated for different deformation levels using the function obtained from *G_eff_* and *A_eff_*.

### 2.2. Mechanical Properties and Selection of the Flexible Reinforcement

The reinforcement provides the necessary vertical behavior for load transmission within seismic isolation devices. This study proposes the evaluation of eight materials to determine their mechanical properties and to identify the most suitable reinforcement for use in seismic isolators. Importantly, all evaluated materials are readily available in the market, ensuring practical applicability and ease of procurement for implementation purposes. These materials, as depicted in Figure 1, include High Modulus Fiber Polyester (HRA), Polypropylene Fiber (TR), High Tenacy Polyester with copolymer 750 (GM), Polyester Mesh Reinforcement 300 (GO), Glass fiber mesh with polyester resin (FP), Glass fiber mesh with epoxy resin (FVE), natural fique fiber (FB), and polyester fiber woven fabric developed by Losanno et al. [21] with epoxy resin (PLE).

Given that the modulus of elasticity, thickness, and Poisson’s ratio are the characteristics reflected in the vertical stiffness of the isolator, these were determined through measurements and mechanical tests on samples of the reinforcements, as mentioned earlier.

Tensile tests were conducted at the Materials Laboratory of Pontificia Universidad Javeriana Cali (PUJC), following ISO 527-4 [24] standards. Five specimens of each material were prepared, measuring 25 mm in width by 250 mm in length. The thickness of each specimen was measured at three points: the right end, left end, and center, using a digital caliper with an accuracy of ±0.01 mm. The average was computed, and this thickness was used for subsequent processes (refer to Table 1).

Subsequently, the specimens were individually clamped in an INSTRON machine with a maximum load capacity of 10 kN, located at the PUJC. Throughout the test, load and deformation data were recorded to calculate the modulus of elasticity and ultimate tensile strength.

Stress and deformation data were recorded during the tensile tests for each fiber. The strain was controlled using an extensometer measuring up to 0.00001 mm/mm in a length of 50 mm, as presented in Figure 2a. The standards stipulate that the material’s modulus of elasticity should be determined using the slope between the strain range of 0.0005 and 0.0025 for each type of tested fiber (HRA, TR, GM, GO, VE, FP, FVE, FB, and PLE). The obtained results are depicted in Figure 2b, which shows the graph representing the average results for each reinforcement tested.

Different materials were evaluated to obtain their modulus of elasticity, an important parameter that defines the vertical stiffness of the device. It is worth noting that the value of the modulus of elasticity varied between 1% and 10%, with the lowest for FP and the highest for HRA. The HRA fiber exhibited a standard deviation of 71.98 MPa, the highest value, due to the fiber-cutting process. During the preparation for testing, some of its lateral threads frayed because of the fiber’s weave. In contrast, the other fiber specimens showed a standard deviation ranging from 1% to 6% in the modulus of elasticity, resulting in a small data dispersion.

Upon analyzing the mechanical performance of the evaluated materials, as detailed in Table 1, FVE demonstrated a superior modulus of elasticity compared to the other materials, outperforming the average of the other materials by 54%. When evaluated for a reduced thickness, its elasticity modulus reached 2280 MPa, representing a 94% improvement compared to the reinforcement (PLE) used by Losanno et al. [21]. In contrast, when examining the other materials, it was observed that, despite having high tensile strengths, they showed significantly low values of elasticity modulus, accompanied by low tensile stress, as seen when examining the product E*Tf of the FVE, recorded in N/mm, as shown in the calculation of vertical stiffness in Equation (1). Table 1 presents this product to ensure it yields an optimal value that increases vertical stiffness. A higher value of this product results in a smaller value of the parameter $\alpha^{2}$, as shown in Equation (2), which increases the value of $E_{f}^{c}$ and, consequently, the vertical stiffness, as demonstrated by Lossano et al. [21]. Thus, selecting FVE establishes that the device’s stiffness would increase, showing an improvement of 139.8% compared to the reinforcement used by Lossano et al. [17]. It is noteworthy that although the FP material had a higher value of E*Tf than the FVE, it exhibited brittle behavior, with a deformation of only 2%. Additionally, its cutting and fabrication required considerable time and detail. This is why FVE was chosen as the preferred reinforcement material due to its superior mechanical properties. Additionally, its selection was influenced by the ease of acquisition and production, as well as its cost-effectiveness, since the price of the fiber developed by Losanno et al. [21] was reduced by 35.19%. These factors collectively position FVE as an optimal choice for enhancing the performance of seismic isolation devices.

### 2.3. Mechanical Properties and Selection of the Adhesive for Interface of Recycled Rubber Reinforcement

Seismic isolators act as buffers to diminish the destructive forces of earthquakes on structures by decoupling a building from ground movements. A component of these isolators is the adhesive within the interface layer. This adhesive not only maintains the structural integrity of the isolator but also markedly dictates its seismic performance. With the evolution of infrastructure and the construction of taller buildings, the academic and technical scrutiny towards these adhesives has intensified. Their rheological properties, long-term durability, and adaptability of adhesives under diverse load conditions are of growing interest, especially as they play an essential role in the isolator’s effectiveness in mitigating seismic forces [25].

To ensure an optimal bond between the matrix of recycled rubber and the reinforcement in a seismic isolator, it is imperative to select the appropriate adhesive, especially considering the inherent limitations of recycled rubber. Indeed, while natural rubber benefits from the vulcanization process to enhance its adherence, recycled rubber cannot undergo vulcanization, again due to its particulate nature and prior activation of its sulfur bridges. Consequently, the emphasis is placed on the application of resins or adhesives to establish the requisite adhesive connection. The selected alternatives for this purpose include adhesives based on methyl acetate (S), 2-Methylpentane-based adhesives (M), adhesives based on Ethyl 2-Cyanoacrylate (CA), Ethylene Vinyl Acetate adhesives (N), Cyanoacrylate Ester-based adhesives (L), 2-Component Epoxy adhesives (TP), DMI Prepolymer adhesives (AA), and Hybrid Mounting adhesives (PF).

#### 2.3.1. Adhesion Shear Test

Given that the device is subjected to shear forces between the matrix and the reinforcement, it is essential to assess their adhesion by the guidelines outlined in the EN 1465 standard using the adhesion shear test (ST) [26]. To achieve this, five specimens sized 100 mm × 25 mm were manufactured, composed of recycled rubber and the selected reinforcement (FVE), as shown in Figure 3a. These specimens were bonded using adhesives over 12.5 mm to identify which adhesive provides maximum adherence against shear movements. Subsequently, the samples underwent a tensile test at a constant rate, ensuring that the total test duration did not exceed 60 s, to determine the adhesive effectiveness of each compound.

#### 2.3.2. Adhesion Tensile Test

The procedure outlined in ISO 36 [27], called the Adhesion tensile test (TT), was employed to assess the adhesive strength. Three specimens, each with a width of 25 mm and a minimum length of 100 mm, were fabricated using recycled rubber and FVE, as presented in Figure 3b for each adhesive. The tensile load was applied using an Instron machine with a capacity of 10 kN, at a constant rate of 50 mm/min. This approach aims to determine the maximum adhesive stress for each adhesive, enabling the identification of the one with the highest capacity. This standardized testing procedure ensures a comprehensive evaluation of adhesive performance and adherence in a controlled and replicable manner.

#### 2.3.3. Mechanical Properties and Selection of Adhesives

To conduct the TT and ST tests, it was necessary to manufacture the chosen reinforcement, FVE. Simultaneously, the production of recycled rubber was undertaken, involving the creation of a specific mold that allowed the fabrication of specimens following the specifications outlined in ISO 36 [27] and EN 1465 [26] standards, as shown in Figure 3. This process ensures the precision and compliance of the obtained specimens with the established testing norms, facilitating a rigorous and standardized evaluation of the materials.

The adhesion tests (TT and ST) were conducted at PUJC. Each specimen was placed on an Instron machine with a capacity of 10 kN to apply a constant deformation, following ISO 36 [27] and EN 1465 [26] regulations. Force data were recorded to calculate the adhesive capacity of each adhesive, as shown in Table 2. Furthermore, to classify the type of failure, the classifications described in Section 12.2 of the ISO 36 [27] standard were used, which are as follows: RT indicates instances where separation occurred between the elastomer and the reinforcement due to an absence of adhesion; R denotes cases where the rupture originated within the elastomer layer; and RA refers to separations between the elastomer layer and the adhesive [27], as shown in Figure 4.

Adhesives S, M, and N demonstrated an adhesive strength of 0.119 N/mm, which is a lower strength compared to the other adhesives evaluated, experiencing failure between the two analyzed surfaces (RT and RA failure types are shown in Figure 4). On the other hand, adhesives CA, L, T, and AA showed excellent performance, with the matrix failing before the adhesion between surfaces (R-type failure in Figure 4b). However, the economic factors and their presentation were considered when evaluating these adhesives. Given the need for large-scale device production, adhesives L and T do not have a suitable presentation for extensive manufacturing, unlike CA and AA adhesives.

To test the PF adhesive, the necessary amount was applied cold between the laminated surfaces of recycled rubber and reinforcement FVE. However, as presented in Table 2, its performance was 54% lower than the adhesives with higher performance (CA, L, TP, and AA). A bonding process was then carried out by placing an uncured matrix base and the reinforcement with the adhesive, creating a type of sandwich, which was then pressed and heated for 20 min at 140 °C. When attempting to separate the surfaces, it was not possible, as the adhesive perfectly bonded the two parts together, as shown in Figure 5. Thus, the adhesive transitioned from working in cold conditions to performing better when heat was applied in the fabrication of a prototype, allowing the rubber particles to better align within the adhesive. It is significant to note that adhesive AA demonstrated improvement through the same process, as depicted in Figure 5. Figure 5 highlights the compatibility between the matrix and the adhesive, where the bond between surfaces was exceptionally strong.

Considering the mechanical properties and acquisition cost, three adhesives (CA, AA, and PF) were selected to verify their behavior in a seismic isolator prototype. These adhesives exhibited a significant acquisition cost relative to their form and the performance they deliver, with average adhesion strengths ranging from 1.768 to 2.188 N/mm. Additionally, there was ease in obtaining materials from the national market. It is worth noting that adhesives S, M, and TP exhibited a higher standard deviation than the other adhesives, because the glue slid over the surfaces and failed with the RT-type failure, which did not provide adequate adhesion and resulted in lower maximum recorded force. However, other materials, such as CA, when applied to the device, had a deviation of 12% in the shear test results.

## 3. Manufacturing of the Prototype Seismic Isolator

The next step involved performing an applicability test by selecting the optimal adhesives (CA, AA, and PF) and reinforcement based on the evaluation conducted (FVE). This phase included assessing their performance in reduced-scale prototypes of seismic isolators in accordance with Losanno et al. [21], aiming to confirm the effectiveness of the adhesives and reinforcement both individually and in their integrated behavior within a context representative of actual structural conditions.

### 3.1. Monolithic Manufacturing Process

The procedure outlined by Ortega et al. [19] and Madera Sierra et al. [18] was followed for manufacturing recycled rubber prototypes, utilizing the recycled rubber matrix with a dosage and density of 0.99 g/cm^3^. The material was sourced from Occidental de Cauchos SAS. The fabricated prototypes followed the recommendations by Losanno et al. [21], adhering to FEMA 450 requirements, considering a steel structure with four supports with two degrees of freedom and a reduced mass of 1/6, giving an axial load value of 19 kN per column in the building. For the design period, 1.15 s was chosen, using seven ground motions with a return period of 1900 years. Furthermore, the maximum displacement of the structure was calculated at 30 mm for 100% deformation, with a target damping of 10%. Accordingly, prototypes with an 80 mm diameter and 44 mm height were developed, incorporating the selected reinforcement and adhesive, as depicted in Figure 6.

The manufacturing of monolithic seismic isolator prototypes was carried out through a detailed and precise process. A steel mold was specifically designed to house the rubber mixture prepared beforehand, allowing for the strategic interleaving of reinforcement layers and the application of the three selected adhesives: Cyanoacrylate (CA), Acrylic Adhesive (AA), and Phenol Formaldehyde (PF). Each adhesive was tested across different prototypes as detailed in Table 3. This procedure resulted in a structure comprising 15 rubber layers and 14 reinforcement layers, forming an assembly that accurately represents the planned configuration for the seismic isolators.

These layers were consolidated through a pressing process, where the molded assembly was subjected to a specific pressure for 40 min. This critical process was conducted at a controlled temperature of 140 °C to induce adequate adhesion between the recycled rubber particles. The temperature and pressing duration were optimized to allow the recycled rubber agglomerate to fuse appropriately, ensuring the structural cohesion of the prototype, as depicted in Figure 6a,b.

Following the outlined procedure, the manufacturing of the prototypes was carried out and the nomenclature established, considering various combinations of reinforcements, adhesives, and rubber blends. Two reinforcements were selected for this evaluation: the one developed by Madera Sierra et al. [18] (P) and the one set in the present project, based on fiberglass mesh with epoxy resin (FVE).

Additionally, reinforcement sheets with five holes, each measuring 5 mm in diameter, and without holes were considered, as depicted in Figure 6c,d. This approach aimed to determine whether the presence of holes would enhance the adhesion between the rubber layers and reinforcement and to assess their impact on both vertical and horizontal stiffness.

The nomenclature adopted for the prototypes adheres to the format Adhesive/Blend and Reinforcement/Number, offering a clear and systematic method for identifying each combination evaluated in this study, as illustrated in Table 3. This approach facilitates the creation of two prototypes for each combination, ensuring a comprehensive assessment of their performance.

For the recycled rubber blends, three formulations were taken into account: one developed by Ortega et al. [19] and two by Meza et al. [28]. To distinguish between the blends formulated by Meza et al. [28], labels PAFVE10 and PAFVE15 were assigned, indicating the difference between the two rubber mixtures. For the blend developed by Ortega et al. [19], only the name of the reinforcement was retained to identify it, called REF, as presented in Table 3.

### 3.2. Manufacturing Process by Layers

On the other hand, the manufacturing process for seismic isolators was executed in layers, utilizing a detailed procedure predicated on cold bonding. This method was chosen to reduce the amount of energy consumed in the monolithic creation process. The utilization of heat is necessary for activating the adhesive in the monolithic approach, ensuring the adhesion of all the prototype layers as a cohesive unit. In contrast, the layering process via cold bonding aims to achieve similar levels of adhesion without the extensive energy requirements, thereby presenting an energy-efficient alternative in the fabrication of seismic isolators. The adhesive ethyl 2-cyanoacrylate (CA) was used, as the other two selected adhesives do not provide proper cold adhesion. Each layer of the device was formed individually, alternating sheets of reinforcement and recycled rubber sheets, bonded using the mentioned adhesive, as shown in Figure 7.

The layered manufacturing involved creating each in a mold with a diameter of 80 mm and a thickness of 2 mm. Subsequently, the resulting composition was subjected to a temperature of 140 °C for 8 min. This specific temperature allowed the binder to promote adhesion among the recycled rubber particles. Once each sheet of recycled rubber is prepared, the reinforcement and the rubber sheet are bonded together using the cold adhesive CA, resulting in prototypes as depicted in Figure 7.

This construction method, focused on cold bonding, involved a more time-consuming manufacturing process; however, it ensured adequate structural cohesion of the prototype. The result was a stratified device that reflects the conceptual configuration and, through this technique, allows a practical evaluation of the seismic isolator’s performance under conditions close to reality. The production of the prototypes shown was only carried out with the FVE reinforcement and the blend developed by Ortega et al. [19], and with a label denominated CA-PFVE Layers, as presented in Table 3.

## 4. Mechanical Properties of Seismic Isolator Prototypes

### 4.1. Performance Evaluation in the Vertical Direction

In order to evaluate the mechanical behavior under compression and determine the vertical stiffness of the prototypes, a cyclic compression test was conducted in the PUJC using a machine supplied by Geotest Instrument Corp., with a compression capacity of 5 tons; see Figure 8a.

This test consisted of three stages to measure the displacements generated under a variable compression load. Initially, the prototypes endured a compression load applied at a speed of 0.01 mm/s until reaching the design load of 19 kN, as reported by Madera Sierra et al. [18]. Subsequently, three consecutive cycles were used, until reaching ±30% of the design load at a speed of 0.05 mm/s. Finally, the compression load was removed at a speed of 0.01 mm/s, as shown in Figure 8b.

The graphs obtained in the cyclic test determined each cycle’s maximum and minimum values, where the resulting slope reveals the experimental vertical stiffness. It is important to note that the analysis considered the average of the two prototypes tested for each combination of adhesive, recycled rubber, and reinforcement. The combinations of the tested prototypes are presented in Table 3.

### 4.2. Performance Evaluation in the Horizontal Direction

To determine the horizontal response of the prototypes, a cyclic shear test was performed at Purdue University’s Intelligent Infrastructure Systems Laboratory (IISL). In a small-scale setup, a vertical actuator with a capacity of 294 kN maintained a constant vertical force of 22 kN, representing the weight of the superstructure. The vertical load was measured using a load cell transducer with a rated capacity of 88.9 kN, placed between the top plate and the hydraulic jack. The vertical displacement of the top plate was measured by an LVDT (Linear Variable Differential Transformer) transducer with a stroke of ±76.2 mm. A linear, double-ended servo-hydraulic actuator (Shore Western, 910D series), with a maximum rated force of 9.78 kN and a stroke of ±60 mm, was used to apply horizontal displacement to the specimen. An LVDT transducer, built into the actuator, measured the displacement of the actuator’s rod [29]. The displacement protocol was applied, adhering to the recommendations of Losanno et al. [17], involving seven earthquakes in the North–South direction and seven in the East–West direction, selected by ASCE7-16 standards, with a return period of 100 years. The protocol was executed at six deformation levels, 25%, 50%, 67%, 100%, 100%, and 75% of the maximum deformation, which was recorded at 29 mm, as shown in Figure 9a.

The developed prototypes were individually tested within the loading frame, as seen in Figure 9b, where force and displacement data were collected. This process facilitated the determination of horizontal stiffness and damping as provided by the device, as well as its maximum deformation, to evaluate the prototypes’ capacity to endure and mitigate seismic forces, ensuring their efficacy as seismic isolation devices.

## 5. Results and Discussion for Seismic Isolator Prototypes

### 5.1. Cyclic Compression Test

Based on the results presented in Table 4, it was observed that the fiberglass mesh with epoxy resin (FVE) exhibited the anticipated behavior. Compared to the reinforcement used by Ortega et al. [19], there was an improvement of 288.98% in experimental vertical stiffness (Kvexp), with an increase from a value of 17.69 kN/mm to an average of 45.90 kN/mm across all tested prototypes. Additionally, the comparison between theoretical (Kvth) and experimental vertical stiffness (kvexp) revealed a maximum variation percentage of 12.5%, a value lower than the 33.5% reported by Ortega et al. [19]. This discrepancy was attributed to unusual movements of the testing machine, causing the prototype to shift during the test. Furthermore, variations such as 18.2% were higher but represented experimental values exceeding the expected theoretical figures.

Although the reinforcement was perforated with five holes of 5 mm diameter, these modifications did not impact the vertical stiffness. The variation observed was due to the reorganization of rubber particles and the shear modulus of the different mixes used in the prototypes. The selected reinforcement exhibited a maximum residual deformation percentage of 0.68% and a minimum of 0.23%, values lower than those obtained by Losanno et al. [21] and Ortega et al. [19], representing an improvement of 18% and 24% respectively. Notably, the prototypes did not show any damage up to a 30% increase in the applied design load (19 kN). Similarly, as noted in Table 4, the CA-PFVESH and CA-PFVE prototypes exhibited a standard deviation higher than the other prototypes in their vertical stiffness, of approximately 37%. This significant difference is due to the inherent behavior of the rubber material, which is porous and particulate and readjusts under compression loads, as also reported by Ortega et al. [19]. However, the other prototypes showed behavior with small dispersion, varying from 1% to 18%, considering the porous conditions and particle rearrangement of the matrix, as reported by Ortega et al. [19] and Losanno et al. [30].

Additionally, the prototypes fabricated in layers (CA-PFVE Layers) also exhibited an improvement in vertical stiffness compared to the monolithic prototypes developed by Ortega et al. [19], with an increase of 121.48%. This enhancement enabled them to support a greater load with lesser residual deformations, as presented in Table 4. A notable improvement in residual deformation of 36.21% was achieved, allowing structures employing these devices to have a reduced probability of experiencing differential settlements at the supports.

### 5.2. Cyclic Shear Test

When the shear test was conducted, most of the prototypes exhibited a sliding condition on the surfaces. This was due to the low abrasiveness of the recycled rubber, causing the bottom part of the prototypes to wear down, destroying the first layer of rubber, as shown in Figure 10a.

Additionally, when creating monolithic prototypes with the CA adhesive, labeled as CA-PFVE, which does not require a heat curing process, a failure due to lack of adhesion (AF) at the base was observed in these specimens (Table 5). During the shear cycles, the layers detached, as seen in Figure 10b, and the specimen failed to meet the established protocol when the 25% portion of the deformation protocol started, as shown in Table 5. This detachment occurred because, upon contact with heat in the manufacturing monolithic process, the adhesive’ mechanical properties were degraded.

The prototypes designated as AA-PAV15 and CA-PFVESH also exhibited low performance, only achieving a deformation of 25% with a displacement of merely 6.9 mm. This occurred because, during the execution of the test at deformations greater than 25%, the device failed due to friction, as shown in Table 5 (FF), preventing further deformation of the device and thus not allowing energy dissipation.

Considering that the recycled rubber mix used in the PF-FVEH prototypes was the same as that in the CA-PFVESH prototypes, these prototypes, although only achieving a deformation of 50%, demonstrated, as seen in Table 5, that by changing the adhesive from a CA to a PF, which is a flexible adhesive, there was an improvement in damping of 29.46% compared to the CA-PFVESH and an improvement in displacement of 106.35%. Furthermore, in comparison to the prototype developed by Ortega et al. [19], referred to in this research as REF, there was a decrease in horizontal stiffness of 2.5% and an improvement in displacement of 2.3%.

The improvement brought about by the flexible PF adhesive was also demonstrated with the 15% mix developed by Meza et al. [28]; the PF-PAVF15 prototypes, compared to the AA-PAFV15, showed an improvement in horizontal deformation of 5.07% at a deformation less than the 50% at which the AA-PAFV15 prototypes failed due to friction (FF), as well as a decrease in horizontal stiffness of 33.52%.

On the other hand, the prototypes constructed in layers, named CA-PFVE Layers, exhibited better behavior than the prototype presented by Ortega et al. [19]. As illustrated in Table 5, the prototypes achieved a displacement of 18.73 mm at 67% deformation, representing an improvement of 33.98% and an increase in damping of 64.42%. This marks a progression from 17.02% damping at 50% deformation to achieving a higher deformation of 67% with a damping rate of 20.75%, compared to the prototype developed by Ortega et al. [19] and the fiber developed by Losanno et al. [21], as presented in Table 5 (REF). Regarding horizontal stiffness, a significant reduction was observed, moving from a deformation of only 50% with a value of 365.03 N/mm to a higher deformation of 67% with a value of 252.89 N/mm. This represents a reduction in horizontal stiffness of 52.93%, as observed in Table 5. In addition, these prototypes demonstrated the rollover condition of overturning at a deformation of 67%, a condition presented in the prototypes of Losanno et al. [21], as shown in Figure 11a. This behavior did not occur in the prototypes by Ortega et al. [19].

Similarly, the prototypes named PF-PAFV15 exhibited better performance than both the CA-PFVE Layers and those developed by Ortega et al. [19]. In terms of displacement, they completed the entire test protocol, achieving a displacement of 29 mm at 100% deformation, which represents a 93.3% improvement compared to other research [19], as shown in Table 5. Additionally, at 100% deformation, horizontal stiffness was reduced by 67%, providing a damping of 23%, which is a 109% improvement compared to Ortega et al. [19], as observed in Table 5. Moreover, the rollover condition sought for this type of device in a disconnected state [31] was also achieved, as presented in Figure 11b. Regarding Losanno et al. [17], although the PF-PAFV15 prototypes exhibited greater horizontal stiffness due to the properties of the recycled rubber, at the same percentage of deformation (100%), the damping provided by the recycled rubber device was 23%, an improvement of 53.33% over that reported for natural rubber with polyester fiber (15%).

The performance provided by the PF-PAFV15 prototypes demonstrated that by using a flexible adhesive, the recycled rubber mix developed by Meza et al. [28] at 15%, a mix with better properties than that developed by Ortega et al. [19] and the reinforcement of fiberglass mesh with epoxy resin (FVE), it was possible to achieve a 100% deformation of the device.

CA-PFVE Layers and PF-PAFV15 prototypes exhibited optimal performance, aligning with the design principles for low-rise buildings [17]. This enabled the isolators designed (PF-PAFV15) in this study to provide a damping of 23%, a value that is 130% higher than the target (10%). Consequently, this permits the use of the device, since it offers the requisite mechanical properties for its application. Similarly, the prototypes designed in layers (CA-PFVE Layers) but utilizing an adhesive that did not require heat activation (CA) demonstrated the ability to provide damping of 20.75% at a deformation of 65%, also proving to be a promising device for low-rise buildings.

## 6. Conclusions

This paper presents the results obtained when the mechanical properties of seismic isolators are enhanced with a matrix of recycled rubber through the modification of reinforcement and adhesive. By conducting mechanical tensile and adhesion tests, it was possible to identify the optimal materials for reinforcement and adhesive in prototypes of recycled rubber seismic isolators. The goal was to produce devices that are more cost-effective and lighter, particularly for low-rise structures in developing countries. The results demonstrated that by using a fiberglass mesh with epoxy resin, an elasticity modulus of 2280 MPa was achieved. This result was due to the combination of the two elements producing a very thin sheet of 1.35 mm, which, compared to other reinforcement materials, exhibited superior performance in terms of cost, ease of acquisition, and mechanical properties obtained. Compared to the reinforcement by other researchers, there was a 94% improvement in mechanical properties. Similarly, the selected adhesives, one cold (CA) and the other requiring heat for activation (PF), achieved optimal adhesion with the surfaces of the designed prototype. This performance was due to their high compatibility for adhering porous material, such as recycled rubber, to the resin surfaces of the FVE reinforcement.

The findings from this study indicated a significant enhancement in the vertical response of the devices, with the chosen reinforcement improving this response by 288.8%. Furthermore, the construction process itself demonstrated the capability of achieving greater deformations. By employing the same mix as Ortega et al. [19] but changing the reinforcement and adhesive, and opting for a layered rather than a monolithic construction, the deformation was increased from 50% to 67%. Similarly, the maximum damping was enhanced from 17.02% to 20.75% and enabled the rollover condition, underlining the effectiveness of the material and methodological innovations.

Furthermore, the study demonstrated that the type of adhesive used for bonding the surfaces of recycled rubber significantly influences the horizontal response of the devices. By utilizing a flexible adhesive (PF), an improvement in the mechanical properties of the devices was observed. This led to higher deformation and damping percentages, along with a decrease in horizontal stiffness, thereby facilitating rollover in the PF-PAVF15 prototype and effective energy dissipation through this mechanism. The layered prototype, similarly, reinforced and utilizing a cold process adhesive, also showed promise, underlining its potential for seismic isolation applications akin to PF-PAVF15, a device made of recycled rubber with a 15% binder, as developed by Meza A et al. [28], and reinforced with fiberglass mesh and epoxy resin (FVE), bonded with a flexible adhesive (PF).

This approach was proven to significantly enhance the mechanical properties, akin to the layered configuration’s effectiveness. The collective findings illustrate the profound influence of adhesive selection, reinforcement integration, and the strategic use of layered construction in advancing the field of seismic isolation devices.

## Figures and Tables

**Figure 1 polymers-16-02258-f001:**
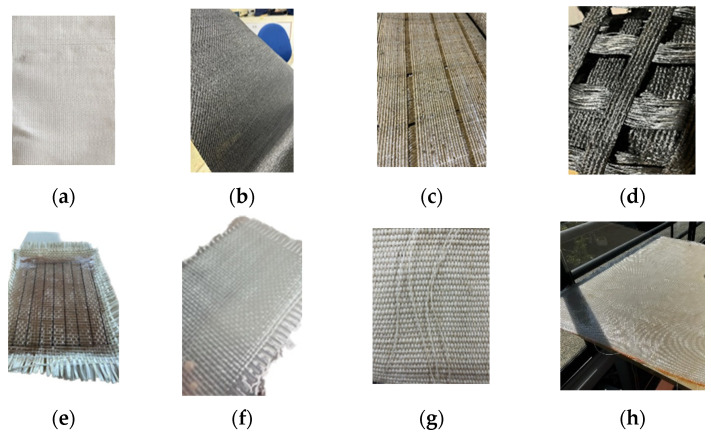
Reinforcements: (**a**) HRA, (**b**) TR, (**c**) GM, (**d**) GO, (**e**) FP, (**f**) FVE, (**g**) FB, and (**h**) PLE.

**Figure 2 polymers-16-02258-f002:**
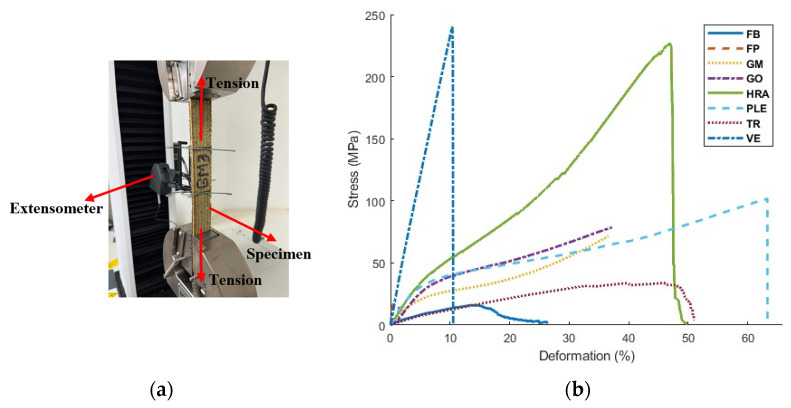
(**a**) Tensile test and (**b**) stress-deformation graphs of reinforcements.

**Figure 3 polymers-16-02258-f003:**
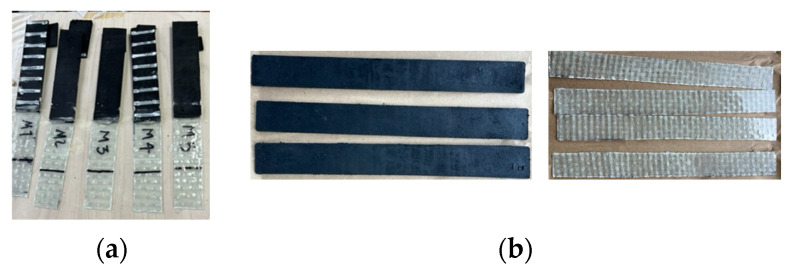
Specimens for (**a**) adhesion shear test and (**b**) adhesion tensile test.

**Figure 4 polymers-16-02258-f004:**
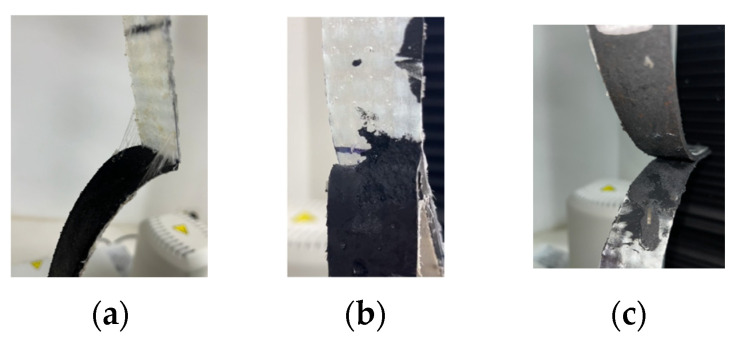
Failure types: (**a**) RT, (**b**) R, and (**c**) RA.

**Figure 5 polymers-16-02258-f005:**
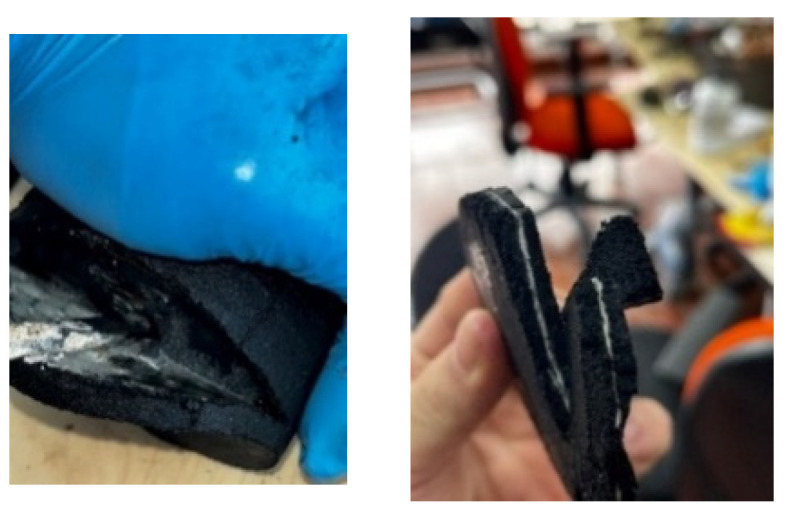
Monolithic adhesion under heat pressure.

**Figure 6 polymers-16-02258-f006:**
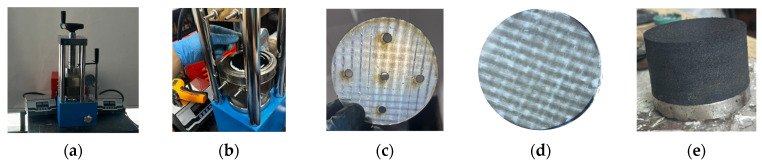
Manufacturing process: (**a**) heat press, (**b**) rubber mold casting, (**c**) reinforcement with holes, (**d**) reinforcement without holes, and (**e**) final product.

**Figure 7 polymers-16-02258-f007:**
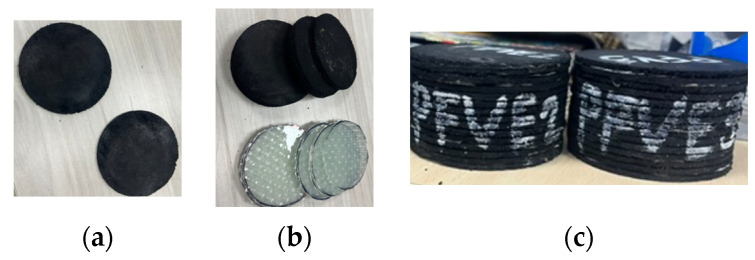
Prototype manufacturing process by layers: (**a**) recycled rubber layer, (**b**) recycled rubber layer and reinforcement FVE without holes, and (**c**) final prototype by layers.

**Figure 8 polymers-16-02258-f008:**
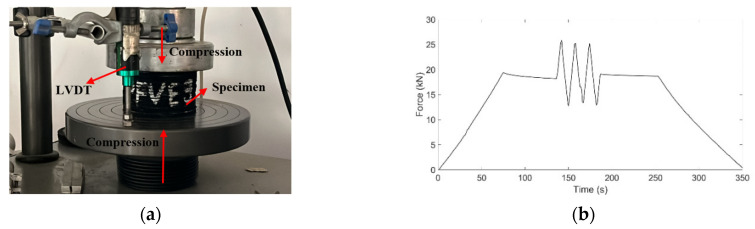
(**a**) Compression test specimen and (**b**) cyclic compression protocol.

**Figure 9 polymers-16-02258-f009:**
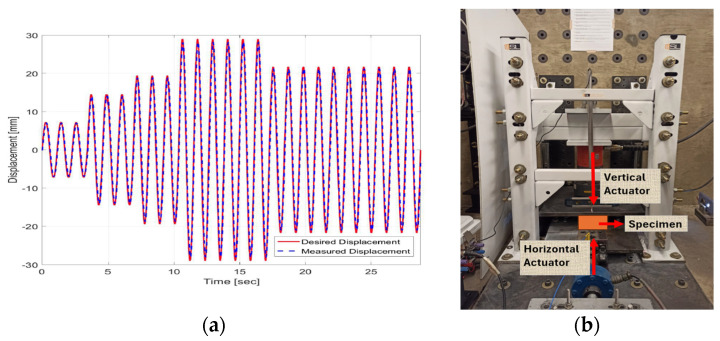
(**a**) Cyclic shear protocol and (**b**) assembly cyclic shear test.

**Figure 10 polymers-16-02258-f010:**
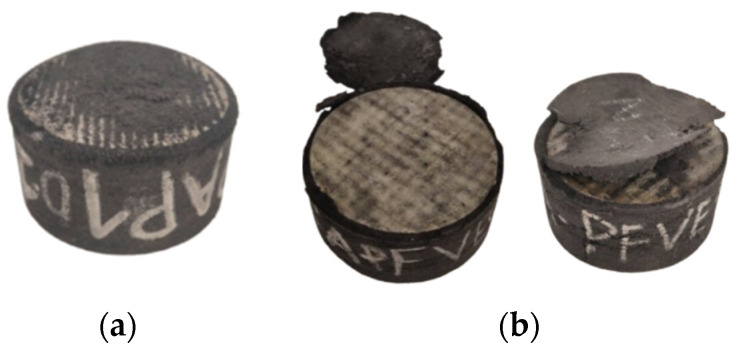
(**a**) Failure due to friction at the base (FF) and (**b**) failure due to adhesion (AF).

**Figure 11 polymers-16-02258-f011:**
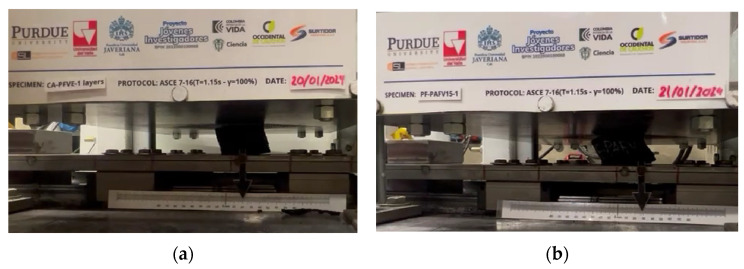
Rollover condition in (**a**) CA-PFVE LAYERS and (**b**) PF-PAFVE15.

**Table 1 polymers-16-02258-t001:** Results of mechanical characterization of reinforcements.

ID	Thickness (mm)	Elasticity Modulus (MPa)		Stress (Mpa)	Deformation	Tf∗E (N/mm)	Poisson Ratio	Cost (USD/m^2^)
		Value	SD					
HRA	0.83	712.27	71.78	263.30	57%	591.18	0.30	8.40
TR	1.1	148.44	9.27	34.00	46%	163.29	0.40	7.50
GM	5.46	567.71	28.41	71.56	38%	3099.69	0.23	19.67
GO	4.85	464.46	20.42	80.68	39%	2252.63	0.20	36.62
FVE	1.36	2279.88	61.02	240.11	11%	3100.63	0.39	17.90
FP	2.52	1432.41	15.12	71.06	2%	3609.67	0.35	15.20
PLE	2.02	691.28	22.15	103.56	63%	1396.38	0.30	27.62
FB	2.24	247.35	4.77	18.68	19%	554.05	0.45	15.25

**Table 2 polymers-16-02258-t002:** Adhesion results.

Adhesive	Presentation	Cost (USD)	Adhesion Strength (N/mm)		Shear Adhesion Strength (N/mm^2^)		Failure Type	Temperature
			Value	SD	Value	SD		
S	500 g	55.98	0.119	0.043	0.142	0.013	RT	Not needed
M	467 g	38.75	0.053	0.018	0.143	0.057	RT	Not needed
CA	28.3 g	15.00	2.188	0.422	1.332	0.161	R	Not needed
N	539 g	45.00	0.089	0.032	0.147	0.011	RT	Not needed
L	5 g	3.25	1.794	1.061	1.710	0.160	R	Not needed
TP	16 g	3.75	0.634	0.444	1.318	0.524	RA	Not needed
AA	1 L	50.00	1.768	0.432	2.527	0.985	R	10 min 140 °C
PF	460 g	17.50	0.541	0.333	0.790	0.257	R	10 min 140 °C

**Table 3 polymers-16-02258-t003:** Labels for specimens prepared for compression and shear cyclic tests.

Matrix	Reinforcement	Adhesive	Holes	Manufacturing Process	Label
Ortega et al. [19]	Polyester	Polyurethane	Not	Monolithic	REF1, REF2
Ortega et al. [19]	FVE	CA	Not	Monolithic	CA-PFVESH1, CA-PFVESH2
Ortega et al. [19]	FVE	CA	Yes	Monolithic	CA-PFVE1, CA-PFVE2
Ortega et al. [19]	FVE	CA	Not	Layers	CA-PFVE1 Layers, CA-PFVE Layers
Ortega et al. [19]	FVE	PF	Yes	Monolithic	PF-PFVEH1, PF-PFVEH2
Meza et al. [28] PA10	FVE	AA	Yes	Monolithic	AA-PAVF10-1, AA-PAVF10-2
Meza et al. [28] PA15	FVE	AA	Yes	Monolithic	AA-PAVF15-1, AA-PAVF15-2
Meza et al. [28] PA10	FVE	PF	Yes	Monolithic	PF-PAVF10-1, PF-PAVF10-2
Meza et al. [28] PA15	FVE	PF	Yes	Monolithic	PF-PAVF15-1, PF-PAVF15-2

**Table 4 polymers-16-02258-t004:** Cyclic compression test results.

Specimen	Kvexp (kN/mm)		Kvth (kN/mm)	Difference	Hi (mm)	Hf (mm)	εres (%)
	Value	SD					
REF	17.69	0.88	20	11.6%	43.99	43.73	0.58%
CA-PFVESH	43.53	16.57	43	1.9%	43.66	43.50	0.37%
CA-PFVE	37.35	13.93	43	12.5%	44.38	44.08	0.68%
CA-PFVE Layers	39.18	9.23	43	8.3%	47.92	47.75	0.37%
PF-PFVEH	50.49	3.12	43	18.2%	44.48	44.33	0.33%
AA-PAVF10	57.29	10.39	49	16.5%	44.99	44.82	0.38%
AA-PAVF15	50.44	0.46	53	5.0%	45.69	45.59	0.23%
PF-PAVF10	43.78	0.22	49	10.9%	45.44	45.28	0.35%
PF-PAVF15	45.11	7.87	53	15.1%	46.27	46.01	0.57%

**Table 5 polymers-16-02258-t005:** Cyclic shear test results.

	Kh (N/mm) for Different Deformation					Damping (%) for Different Deformation					
Specimen	25%	50%	67%	100%	75%	25%	50%	67%	100%	75%	Maximum Displacement (mm)
REF	537.29	365.03	FF	--	--	12.62	17.02	FF	--	--	13.98
CA-PFVESH	528.61	FF	--	--	--	11.20	FF	--	--	--	6.93
CA-PFVE	AF	--	--	--	--	AF	--	--	--	--	AF
CA-PFVE Layers	449.73	317.04	252.89	FF	--	13.93	15.81	20.75	FF	--	18.73
PF-FVEH	524.03	371.09	FF	--	--	12.04	14.50	FF	--	--	14.30
AA-PAVF15	646.02	FF	--	--	--	13.46	FF	--	--	--	6.94
PF-PAVF10	344.45	255.21	212.09	FF	--	12.31	13.20	15.02	FF	--	18.88
PF-PAVF15	429.38	308.68	258.19	188.59	166.26	12.12	13.50	14.46	22.70	22.64	28.40

## Data Availability

Data will be made available on request.

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
