# Peer review of "Alternatives for Enhancing Mechanical Properties of Recycled Rubber Seismic Isolators"

_polymers, 2024, doi:10.3390/polym16162258_

Round 1
Reviewer 1 Report
Comments and Suggestions for Authors
The article presents a very good approach of Alternatives for Enhancing Mechanical Properties of Recycled Rubber Seismic Isolators. It is well written and presented. It can be improved based on the following:
-The abstract should highlight the main numarical findings.
-The Illustration 1. has been mentioned for first 2 figures captions. Also the quality need to be improved.
- Table of comparison of this work and related works in litrature can highlight the main improvment and the contribution of this work.
- The future work and suggsution can be highlighted with conclusion
Comments on the Quality of English LanguageMinor editing of English language required
Author Response
I hope this message finds you well. Thank you in advance for your willingness to review this manuscript. Attached, you will find the document with the requested corrections.
Thank you very much for your attention. We look forward to your comments.
Comment 1: The abstract should highlight the main numerical findings.
Response: The abstract was modified to emphasize the findings.
Comment 2: Illustration 1 has been mentioned for the first 2 figure captions. Also, the quality needs to be improved.
Response: The illustrations were improved and corrected as requested.
Comment 3: A table comparing this work with related works in the literature can highlight the main improvements and contributions of this work.
Response: Although a comparison table is not presented, the contribution of this work is made much clearer in the literature review.
Comment 4: The future work and suggestions can be highlighted in the conclusion.
Response: The conclusions clarify the impact of using seismic isolation devices in developing countries.
Reviewer 2 Report
Comments and Suggestions for Authors
Dear authors,
thank you very much for the article. It is advised that it must be structured in a good form. This will make it easier to understand your work.

Author Response
I hope this message finds you well. Thank you in advance for your willingness to review this manuscript. Attached, you will find the document with the requested corrections.
Thank you very much for your attention. We look forward to your comments.
Comment 1: The selection of literature is appropriate to the topic. Please check the information used in Reference 18. The specified characteristic values cannot be found in the article [18].
Response: The information in the requested reference has been updated.
Comment 2: In principle, it is recommended that this section be restructured to improve readability. The title of this section is "Materials and Procedures". The materials are presented rather incidentally. The results presented on page 7, Figure 3, should not be discussed here but in Section 3, as this makes the article easier to read.
Response: The titles of the document have been changed to specify the procedures developed more clearly, adjusting the figures and tables for better readability.
Comment 3:Line 152: Double designation "Equation 1"
Response:Corrected.
Comment 4: Where do you get the value for the thickness of the rubber layer? In your discussion of results, this value is not discussed at all. Please explain!
Response:The origin of the rubber thickness is explained, referencing the source of comparison and design.
Comment 5: Equation 8: The explanation in the text is completely missing. Please explain!
Response:The use of this equation, now Equation 7, is detailed to clarify the topic and its explanation.
Comment 6: Line 188: The source of this reference is missing: Russo et al.
Response:The source has been incorporated.
Comment 7: Point 2.2: This section describes results and does not match 2. For better readability, you should reorganize this section in the paper.
Response: The titles and their explanations were reorganized for better clarity.
Comment 8: Figure 3b: The standard for the tensile test according to DIN ISO 527 specifies that the elongation must be given in "%". The unit "mm/mm" is already mathematically incorrect. Please change this, also in the tables! Line 244: N per mm is not an SI unit! Please change to N/mm!
Response: The units were adjusted as requested.
Comment 9: Line 252: Please explain the parameter Tf*E!
Response: The parameter was detailed, referring to the performance of the material's thickness and its obtained mechanical properties.
Comment 10: Table 1: For a better classification of the results, it is essential that you state and discuss the standard deviation. This is the only way to determine whether the change in properties is significant.
Response:The standard deviation for each evaluated material was explained and shown.
Comment 11: Section 2.1.1: It's really hard to read and follow you when you give references to pictures that are only shown pages later.
Response: The figures were reorganized for better clarity.
Comment 12-13-14-15: Line 295: Please check: EN 1645? Where are you shown in Figure 5? Please explain and change!
Line 306: The EN1645 standard is incorrect!
Line 336: The EN1945 standard is incorrect!
Note: The ISO 36 standard states that the fracture designation should be in accordance with EN ISO 10365. It is therefore not necessary for you to come up with your own fracture designations. Please use the designations from the standard!
Response: The standard was updated. There was a typo, and the standard is EN 1465. The failure modes were detailed exactly as specified in section 12 of ISO 36.
Comment 16: Line 432: The position of Table 3 does not fit here at all. Unfortunately, it makes it very difficult to read and understand your work.
Response: The table was reorganized for better clarity in the text.
Comment 17: Line 470: This sentence is incomplete!
Response: The sentence was completed, and the idea was clarified.
Comment 18: Line 526: What do the parameters Kvexp and Kvth mean? Please explain! For a better understanding of the values, you should definitely state the standard deviation of your measured values and also include it in the discussion.
Response: The acronyms were clarified in the text, and the standard deviation for these procedures was presented.
Round 2
Reviewer 2 Report
Comments and Suggestions for Authors
Dear authors,
The paper has clearly been revised. This has led to a significant improvement in readability. However, minor changes can still improve this.

Author Response
I hope this message finds you well. Thank you in advance for your time and willingness to review the manuscript. We have made the requested corrections to the document.
Thank you for your attention throughout this process. Below, I detail the actions taken based on your proposed recommendations:
Comment 1: Line 153: You wrote: "Equation Equation (1)" - duplication! Please correct!
Response: Corrected.
Comment 2: Line 203: Reference error is displayed in the paper! Please correct!
Response: Corrected.
Comment 3: Table 1: You can delete the unit for the deformation of the mean values if you have already specified the unit in the table header. SD (MPa) - which parameter are you discussing the standard deviation for?
Response: The table has been revised to clarify the parameter for which the standard deviation is determined. Additionally, a discussion of these parameters has been included in the text.
Comment 4: Line 249: Unfortunately, the meaning of the sentence is difficult to understand. The quantity Tf*E has not yet been clearly named. Where does this value come from and what does it indicate about the material?
Response: The description has been clarified to explain the purpose of this parameter and its role in the development of the seismic isolation device.
Comment 5: Table 2: The index at N/mm² must be displayed in superscript. What standard deviation (SD) are you discussing here?
Response: The table has been revised for clarity regarding the standard deviation parameter, and a discussion of these parameters has been detailed in the text.
Comment 6: Line 475: You refer to Table 4 (page 13). However, this is only shown at the bottom of page 14. Please change! What standard deviation (SD) are you discussing here?
Response: The location of the table has been corrected, and the parameter related to the standard deviation has been clarified. This deviation has also been discussed in the text.
Comment 7: Line 522: You refer here to Table 5 (page 14), which you then only show on page 16! Please change! What standard deviation (SD) are you discussing here?
Response: Corrected.
Best regards,
Faider Sebastian Rivas Ordoñez
